# Tunable colloidal swarmalators with hydrodynamic coupling

Veit-Lorenz Heuthe [1,3], Priyanka Iyer [2,3], Gerhard Gompper [2] & Clemens Bechinger [1] ✉

Swarmalators - entities that combine swarming with synchronization - offer a powerful framework for understanding systems where spatial organization and internal degrees of freedom are bidirectionally coupled. Such interplay arises in diverse natural and engineered systems, from Japanese tree frogs and magnetic domain walls to robotic swarms. In contrast to an established theoretical framework, experimental realizations with tunable coupling between motion and phase remain elusive. Here, we present a controllable swarmalator system based on feedback-controlled self-propelling colloidal particles orbiting around reference points and interacting via hydrodynamic flows. We show that synchronization and spatial dynamics co-evolve, giving rise to collective states including synchronized clusters, rotating aggregates, and dispersive phases using a single control parameter. A rapid change of this parameter between regimes of attractive and repulsive phase-mediated interactions yields dynamic regimes inaccessible to systems with static interactions. Simulations incorporating squirmer and lubrication forces support our findings. We also find a new interaction channel through synchronization-dependent forces perpendicular to the connection axis between swarmalators. In general, our platform provides a versatile testbed for probing swarmalator physics but also offers novel strategies for the design of self-organizing active matter.

Understanding the emergence of collective behavior from local interactions is a central question in the study of complex systems. In nature, large-scale spatio-temporal patterns frequently arise without centralized coordination as seen, e.g., from the coordinated motion in animal groups[1–3] or the synchronized flashing of fireflies[4,5], chirping of crickets[6] or waving of crabs[7]. Two foundational models—the Vicsek model, which captures spatial organization via local alignment[8], and the Kuramoto model, which describes phase synchronization among coupled oscillators[9]—have provided key insights into such emergent phenomena. However, many real-world systems feature an inseparable interplay between spatial and oscillatory dynamics, rendering these models individually insufficient. Examples include chorusing frogs[10], swimming sperm cells[11], interacting magnetic colloids[12], or magnetic domain walls[13], where agents not only exhibit translational motion but

also internal oscillations which are mutually coupled. To address this type of behavior, the concept of swarmalators (swarming oscillators), i.e., agents that exhibit both collective motion and synchronization, has been proposed[14]. Bidirectional coupling between these modes, where spatial proximity alters phase dynamics and phase differences modify motion, yields rich behaviors such as synchronized clustering, rotating waves, and ring-like assemblies[15,16].

Although several of the above-mentioned systems exhibit phenomenological signatures of swarmalator behavior, the underlying mechanisms linking synchronization to motion in these systems are usually not clear and, accordingly, cannot be controlled. Understanding the coupling mechanisms in more detail is not only crucial to describe experimental swarmalators within existing theoretical frameworks but also important for spatio-temporal control in swarm

[1]Fachbereich Physik, Universität Konstanz, Konstanz, Germany. [2]Theoretical Physics of Living Matter, Institute for Advanced Simulation, Forschungszentrum Jülich, Jülich, Germany. [3]These authors contributed equally: Veit-Lorenz Heuthe, Priyanka Iyer. ✉e-mail: clemens.bechinger@uni-konstanz.de

robotics[17–19], self-assembled micro machines[20–23], and neuromorphic or reservoir computing[24,25].

In this work, we present an experimental realization of a swarmalator system using feedback-controlled self-propelled colloidal particles (denoted as active Brownian particles (ABPs)) with independently tunable spatial and oscillatory dynamics. Active particles are especially well suited for realizing swarmalators, since they pose a comparably easy way of manipulating high numbers of moving agents and because the interactions between active particles have yielded basic swarmalator characteristics before[12,26]. Hydrodynamic interactions at high particle density mediate strong coupling between motion and phase. By tuning a single control parameter, we observe transitions between tightly synchronized, dense clusters and loosely connected, weakly synchronized states, in good agreement with numerical simulations. Our explicit understanding of the coupling mechanism enables us, for the first time, to describe and understand our system within the theoretical framework of swarmalators. Crucially, our system exhibits not only synchronization-dependent attraction, but also lateral interaction forces that drive collective rotation dynamics reminiscent of rotating biological clusters, such as those formed by bacteria or starfish embryos[27,28]. Furthermore, because the timescales for synchronization and clustering of the swarmalators are clearly separated, we can dynamically modulate their interaction strength to access non-equilibrium swarmalator states inaccessible under static conditions. Our results demonstrate a controllable experimental platform for swarmalators and expand the theoretical framework to include previously unexplored physical interactions.

## Results

### Synchronization of oscillating ABPs

To experimentally realize oscillators, we employed ABPs whose propulsion directions are individually controlled by an incident laser beam[29–31]. The ABPs we used were made from carbon-coated silica spheres (radius $a = 3\,\mu m$) suspended in a near-critical water-lutidine mixture. Illumination with a focused laser heats each particle slightly above the critical temperature of the solvent, inducing local demixing and self-propulsion. Crucially, when the laser is laterally offset from the particle center, propulsion occurs opposite to the offset direction (Fig. 1a; for details, see Methods section "Experimental realization of oscillating ABPs"). To encode a phase into each ABP's motion, we programmed them to perform quasi-circular oscillations around designated target positions. This was implemented by continuously steering each particle $i$ in the (memorized) direction of its reference position $\mathbf{q}_i$ using a control signal delayed by one time step $\delta t$ (Fig. 1b; Methods section "Experimental realization of oscillating ABPs"). As a consequence of this control scheme, each ABP performs an orbiting motion around its reference position $\mathbf{q}_i$[32,33] with the oscillating phase defined by $\theta = \tan^{-1}\left(u_y/u_x\right)$ and the spatial components $u_x$ and $u_y$ of the propulsion velocity $\mathbf{u}$. In our experiments, the time delay was set to $\delta t = 0.4\,s$, which is identical to the time interval between recordings of the particle positions. In combination with a propulsion speed of $|\mathbf{u}| \approx 2\,\mu m/s$, this yields in an average oscillation radius of $R \approx 1\,\mu m$ (Fig. 1c). The initial direction of rotation is random for each particle and may spontaneously reverse due to thermal fluctuations. When operating several such oscillators with small distances $|\mathbf{q}_{ij}|$ between their reference points, their oscillation becomes highly synchronized (see Supplementary video 1). Figure 1d shows experimental snapshots of synchronized oscillators with their trajectories colored according to time, where the synchronization is visible in the form of particles moving in the same directions at the same time. Such synchronization is due to hydrodynamic interactions as discussed in detail below. As a measure of synchronization $\sigma_i$ of oscillator $i$ with its neighbors $j$, we use

$$\sigma_i = \frac{1}{N_j} \sum_j \left\langle \cos\left(\theta_i - \theta_j\right) \right\rangle_T. \tag{1}$$

(see Fig. 1e). Here, $\langle \ldots \rangle_T$ denotes the average over time intervals $T = 10\,\delta t$ and $N_j$ the number of neighbors $j$ of oscillator $i$. Accordingly, $\sigma_i$ can vary between +1 and −1, corresponding to in- and out-of-phase oscillations, respectively. Figure 1f shows an array of about 250 oscillators, with their reference positions arranged on a hexagonal lattice at a spacing of $|\mathbf{q}_{ij}| = 12\,\mu m$. The ABPs are color-coded according to their local synchronization $\sigma_i$. The experiments are complemented by numerical simulations incorporating the same geometry and hydrodynamic interactions between ABPs (see Methods section "Simulation model" and SI section 5). At short inter-oscillator distances, the system exhibits large synchronized domains interspersed with only narrow domain walls of weak coherence. As the spacing $|\mathbf{q}_{ij}|$ between neighboring reference positions increases, the mean synchronization across the array decreases (Fig. 1g; Supplementary video 1). This spatially dependent synchronization is a hallmark of swarmalator systems.

To probe the underlying hydrodynamic coupling mechanism, we measured the velocity of a passive colloid placed near a moving ABP as a function of its relative position (Fig. 1h, i). The resulting strongly anisotropic velocity field and the associated force distribution indicates that hydrodynamic interactions mediate the coupling, consistent with prior observations in colloidal oscillators[11,34,35]. The measured flow field agrees well with a superposition of a squirmer model (with active stress parameter $\beta = 0.25$) and short-range lubrication forces (Fig. 1j; Methods section "Simulation model" and SI section 5). Incorporating this interaction model into simulations quantitatively reproduces the experimentally observed distance-dependent synchronization (Fig. 1g, snapshots in Fig. S1, for more details about the hydrodynamic synchronization see SI section 4).

### Motile oscillators

In addition to phase oscillations, a characteristic feature of swarmalators is a translational degree of freedom, with mutual coupling between position and phase. Experimentally, we implement this by allowing the ABP's reference positions $\mathbf{q}_{i,t}$ to evolve dynamically in time. At each time step $\delta t$, they are updated according to

$$\mathbf{q}_{i,t+\delta t} = \mathbf{q}_{i,t} + \Gamma \cdot \left(\mathbf{R}_{i,t} - \mathbf{q}_{i,t}\right). \tag{2}$$

Here, $\mathbf{d}_{i,t} = \mathbf{R}_{i,t} - \mathbf{q}_{i,t}$ is the displacement between the time-averaged particle position $\mathbf{R}_{i,t} = \frac{1}{T}\sum_{t-T}^{t}\mathbf{r}_{i,t}$ and the reference position $\mathbf{q}_{i,t}$ and $\Gamma$ is a freely adjustable coupling parameter. As a result of this rule, the reference positions acquire a time-varying velocity that depends on the displacement $\mathbf{d}_{i,t}$ and is therefore sensitive to the orbiting dynamics of the ABPs. The averaging window $T$ is chosen to slightly exceed a typical oscillation period (6–8 $\delta t$) and is consistent with that used to compute the synchronization parameter $\sigma_i$ in equation (1); moderate variations in $T$ do not qualitatively affect the behavior of the system. To prevent phoretic clustering of the ABPs, we include a short-range repulsive interaction between neighboring reference positions as usual in swarmalator models (see Methods section "Reference position update").

Figure 2a, b illustrate the dynamics resulting from equation (2): an ABP circles around a motile reference point, which itself moves. The mean squared displacement (MSD) provides a quantitative measure of activity[36], and we use it to characterize the mobility of the swarmalator population and its dependence on the coupling parameter $\Gamma$. The MSD of isolated oscillators for different $\Gamma$ values is shown in Fig. 2c. For $\Gamma = 0$, reference positions remain fixed, and the MSD shows oscillations at short times and eventually saturates. For finite values of $\Gamma$, the reference positions move due to fluctuations in the mean position $\mathbf{R}_{i,t}$ of the ABPs. This leads to an increasing value of the MSDs at large times with a $\Gamma$-dependent exponent $\alpha$. Figure 2d shows that $\alpha$, obtained from fits to simulated MSDs, increases from 0 to 1 with increasing $|\Gamma|$ (for $\Gamma > -0.2$), indicating diffusive motion in the limit of large $\Gamma$. For strong negative coupling ($\Gamma < -0.2$), reference positions outrun the ABPs orbits,

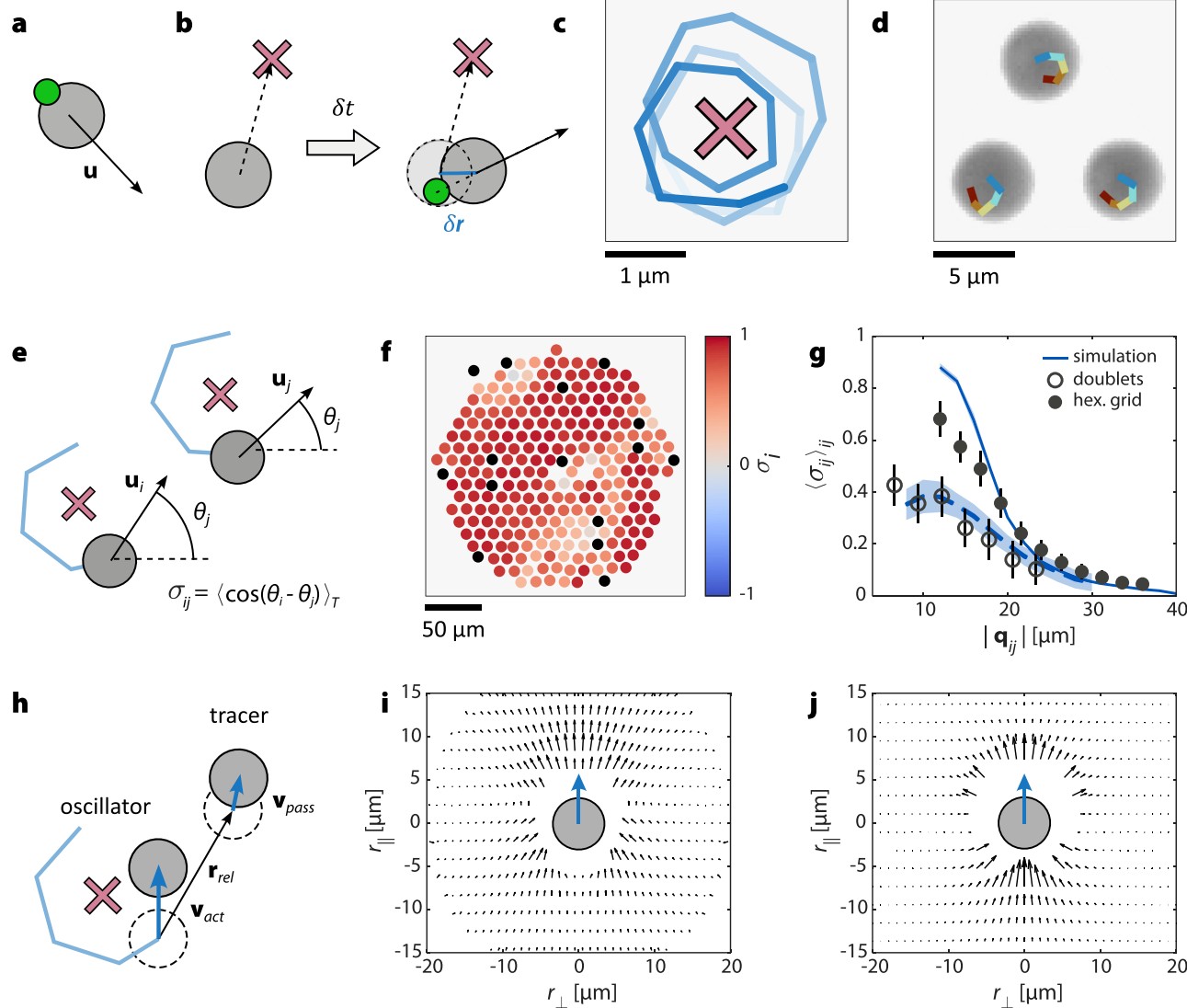

**Fig. 1 | Hydrodynamic synchronization of oscillating ABPs. a** An active colloid (gray, ABP) with velocity **u** propelled by a laser spot (green). **b** An ABP propelled toward a reference position (red) with a time delay $\delta t$ with the use of a laser spot (green). During $\delta t$, the particle moves the distance $\delta r$, and the particle misses the reference site, leading to an oscillation around the reference position. **c** Trajectory (light to dark blue as a function of time) of an oscillating ABP around its reference position (red) from experiments. The ABP position is sampled every $\delta t$, with straight lines connecting successive points to depict the trajectory. **d** Experimental snapshot of three ABPs orbiting around their reference positions in phase. Their trajectories are colored according to time (red to blue). The ABPs are moving in roughly the same direction in each time step, visualizing the synchronization of their oscillation. **e** Definition of the synchronization of two oscillators $i$ and $j$, $\sigma_{ij} = \langle \cos\left(\theta_i - \theta_j\right) \rangle_T$. **f** Experimental snapshot of an array of around 250 oscillators whose reference positions **q** are arranged on a hexagonal grid with a distance $|\mathbf{q}_{ij}| = 12$ μm between neighboring reference positions. The ABPs are colored according to the mean

synchronization to their neighbors $\sigma_i = \langle \sigma_{ij} \rangle_j$ (black denotes defect ABPs). For an animated version, see Supplementary video 1. **g** Average synchronization $\langle \sigma_{ij} \rangle_{ij}$ of oscillators as a function of their reference point distance $|\mathbf{q}_{ij}|$. Closed symbols: 400 oscillators on a grid as shown in (**f**). Open symbols: isolated doublets of oscillators. Blue lines show simulation results based on the force field shown in (**j**), including defect particles (see SI section 3; Fig. S7 for the synchronization without defects). Error bars and light colored areas correspond to the standard deviation. **h** Illustration of the experiment used to map out the forces between ABPs. One oscillating ABP with its target position (red) is placed close to a passive tracer particle, and the velocity of the tracer $\mathbf{u}_{pass}$ is recorded as a function of its position $(r_\perp | r_\parallel)$ in the reference frame of the velocity $\mathbf{u}_{act}$ of the oscillating ABP to determine the hydrodynamic forces. **i** Force field experienced by a passive tracer particle close to an oscillating ABP as a function of its position $(r_\perp | r_\parallel)$ perpendicular and parallel to the ABP. **j** The force field of a combination of a squirmer with $\beta = 0.25$ and lubrication forces, as used in our simulations. Source data are provided as a Source data file.

inducing ballistic motion and quenching of oscillations. Because the ABPs cease to oscillate in this regime, we restrict our analysis to $\Gamma \geq -0.2$ in the subsequent sections.

## Synchronization-dependent swarmalator motion

So far, we have shown that ABP oscillations synchronize as a function of the inter-oscillator distance $|\mathbf{q}_{ij}|$, with reference positions evolving dynamically according to equation (2). What fundamentally distinguishes swarmalators from simple motile oscillators, however, is the bidirectional coupling between phase synchronization and translational

motion. In our system, this coupling arises naturally through hydrodynamic interactions, particularly pronounced at high particle densities. Figure 3a, b shows experimental and simulation snapshots for coupling values $\Gamma = 0.2$ and $\Gamma = -0.1$, respectively. In both cases, systems were initialized with identical ABP densities and evolved for over $6000\delta t$. For $\Gamma = 0.2$, oscillators mutually attract, leading to the emergence of dense, highly synchronized clusters. In contrast, for $\Gamma = -0.1$, this cohesion breaks down: the system expands, the particle density decreases, and synchronization is lost (also see Supplementary video 2).

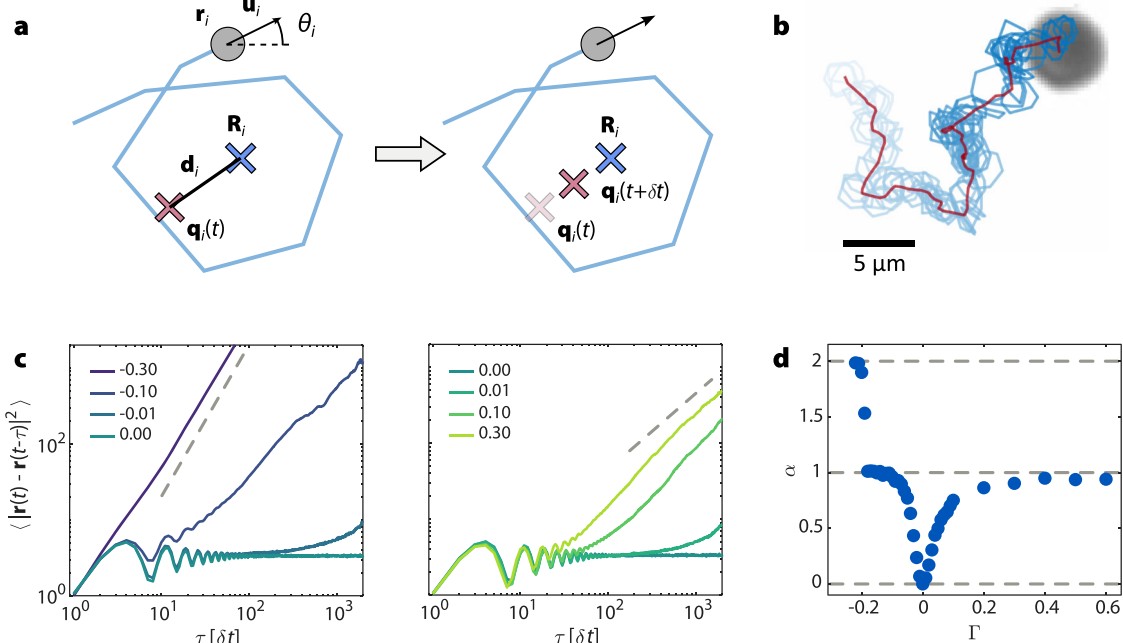

**Fig. 2 | Dynamics of motile oscillators. a** Illustration of a motile oscillator consisting of an ABP (gray) with position $r_i$ and velocity $u_i$ oscillating around its reference position $q_i$ with phase $\theta_i$. The ABP and reference position trajectories are illustrated in blue and red, respectively. According to equation (2), $q_i$ is moved in the direction of the displacement $d_i$ of the ABP position $R_i$ averaged over the interval $T = 10\delta t$. **b** Experimental snapshot of the motion resulting from the update scheme in (**a**) with $\Gamma = -0.1$. The trajectory of the ABP is shown in light to dark blue as a function of time, and that of its reference position in red. **c** Mean squared displacement (MSD) of the motile oscillators from experiments with low particle densities for different values of $\Gamma$ (left: $\Gamma < 0$, right: $\Gamma > 0$, median curves over particles). The dashed gray lines mark $\propto \tau^2$ (left) and $\propto \tau$ (right). See SI section 6 for the definition of the MSD. For $\Gamma = 0$, the MSDs show oscillations as expected for an ABP orbiting around a fixed position. When $|\Gamma|$ increases, the exponent $\alpha$ of the MSD increases, showing a transition from stationary to diffusive ($\alpha = 1$) and ballistic ($\alpha = 2$) motion. **d** Long time exponent $\alpha$ of the MSDs against $\Gamma$ obtained from fitting simulated curves. The dashed lines mark ballistic ($\alpha = 2$), diffusive ($\alpha = 1$) and no motion ($\alpha = 0$). Source data are provided as a Source data file.

Figure 3c shows that the experimentally measured fraction $f_c$ of oscillators within clusters (defined as those with at least one neighbor within $|q_{ij}| \leq 15\,\mu m$ distance) increases continuously with $\Gamma$, consistent with simulations. Notably, isolated oscillators display diffusive motion for both positive and negative $\Gamma$ (Fig. 2d), indicating that the observed structure formation at high densities must arise from interactions between oscillators. Considering the reference point update rule equation (2), this behavior results from the displacement $d_{i,t}$ of oscillating ABPs from their reference positions.

A defining feature of swarmalators is that their motion depends jointly on their distance and phase synchronization. To probe this relationship, we analyzed $d_{i,t}$ as a function of both the reference point separation $|q_{ij}|$ and the pairwise synchronization $\sigma_{ij}$. Figure 3e presents the average longitudinal displacement component $d_{\parallel,ij}$ of $d_{i,t}$, projected along the direction $\hat{q}_{ij}$ to neighboring oscillators. The data reveals a clear dependence of $d_{\parallel,ij}$ on both distance $|q_{ij}|$ and synchronization $\sigma_{ij}$, confirming that motility in our system is jointly modulated by spatial proximity and synchronization—consistent with theoretical swarmalator models. Figure S2 shows a graph of $d_{\parallel,ij}$ against $\sigma_{ij}$ averaged over distance $|q_{ij}|$ between 8 and 15 µm. When two oscillators $i$ and $j$ are negatively synchronized, the longitudinal displacement $d_{\parallel,ij}$ is negative, indicating that each ABP is, on average, displaced away from its neighbor relative to its own reference position. Conversely, for positively synchronized pairs, ABPs are displaced toward each other (i.e., $d_{\parallel,ij} > 0$). The magnitude of this displacement decays with increasing separation between reference positions. This behavior provides a mechanistic explanation for the $\Gamma$-dependent clustering observed in Fig. 3a, b. When two oscillators are synchronized, their ABPs tend to shift toward each other, yielding positive $d_{\parallel,ij}$ values (Fig. 3d). For positive $\Gamma$, this displacement drives the reference positions closer together via equation (2), thereby promoting aggregation. In contrast, for negative $\Gamma$, the reference points move opposite to the

displacement, causing synchronized oscillators to effectively repel. This interpretation is supported by direct measurements of the relative velocities between neighboring reference positions (Fig. 3f), which vary systematically with $\Gamma$, confirming the predicted dependence of effective interactions on the sign of $\Gamma$. Simulations based on the hydrodynamic interaction model detailed in Section "Synchronization of oscillating ABPs" reproduce the experimentally observed trends in both ABP displacement and relative reference-point velocity (Fig. S3). Moreover, the nature of the interaction between oscillators depends sensitively on the character of their hydrodynamic propulsion mechanism: whether they behave as pushers, pullers, or neutral squirmers (Figs. S9, S10). This underscores the critical role of the hydrodynamic flow fields generated by the orbiting ABPs[34,37].

Our results show that the motile oscillators exhibit synchronization-dependent motion and a distance-dependent synchronization. Taken together, these features are hallmarks of swarmalator behavior, consistent with theoretical studies. Standard swarmalator models are characterized by two parameters, the synchronization strength $K$, and the spatial interaction parameter $J$, controlling the synchronization-dependent attraction ($J > 0$) or repulsion ($J < 0$)[14,38,39]. For constant, distance-independent attraction, theory predicts that swarmalators with $K > 0$ and $J > 0$ form a single, dense, synchronized cluster[14]. However, when attraction decays with distance —as in our system—smaller, isolated synchronized clusters are expected[39,40], in agreement with our experimental observations for $\Gamma > 0$ (Fig. 3a). The regime $J < 0$ corresponding to $\Gamma < 0$ is rarely explored theoretically. In our experiments, it leads to dispersed, weakly synchronized swarmalators.

**Swarmalator model**

Beyond a simple qualitative comparison, our quantitative characterization of the interactions among swarmalators enables us to model

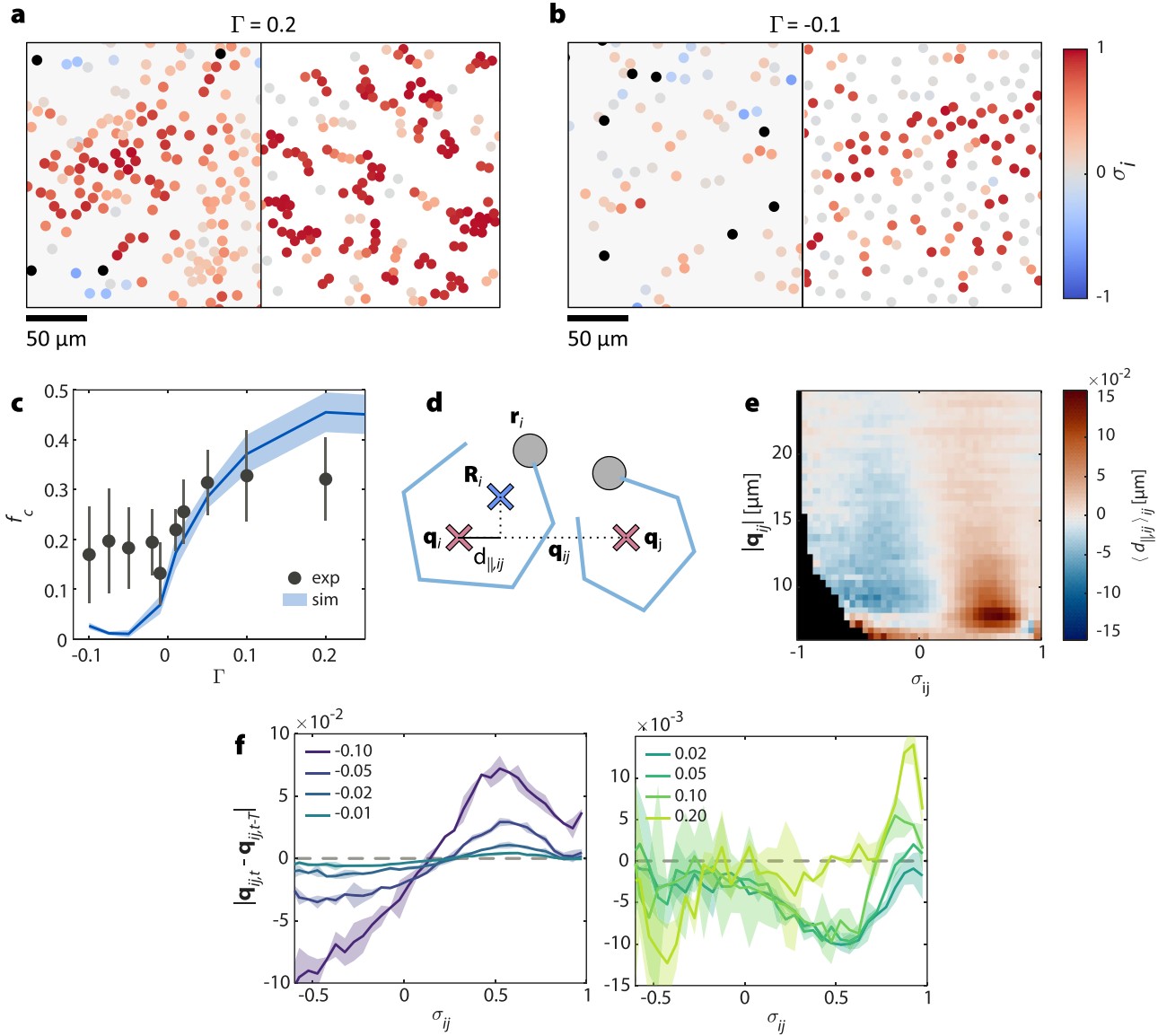

**Fig. 3 | Interactions between swarmalators. a, b** Snapshots from experiment (left) and simulation (right) for $\Gamma = 0.2$ and $\Gamma = -0.1$ (see Supplementary video 2 for animated versions). The swarmalators are colored according to the synchronization averaged over their neighbors $\sigma_i$ (black denotes defect ABPs). With $\Gamma > 0$, the swarmalators form highly synchronized clusters, while for $\Gamma < 0$ the swarmalators disperse and the synchronization is low. **c** Fraction $f_c$ of particles in clusters (with at least one neighbor closer than 15 µm) in experiment (black symbols) and simulation (blue line) against $\Gamma$. The error bars and light colored area correspond to the standard deviation between different runs. **d** A swarmalator with its reference position $\mathbf{q}_i$ (red) and mean particle position $\mathbf{R}_i = \langle \mathbf{r}_i \rangle_T$ (blue). $d_{\parallel, ij}$ is the displacement $\mathbf{d}_i = \mathbf{R}_i - \mathbf{q}_i$ of the ABP from its reference position projected onto the connection vector $\mathbf{q}_{ij} = \mathbf{q}_j - \mathbf{q}_i$ to the reference position $\mathbf{q}_j$ of a neighbor. **e** Heat map of $d_{\parallel, ij}$ averaged over particles against the synchronization $\sigma_{ij}$ with and distance $|\mathbf{q}_{ij}|$ to neighbors. Figure S2 shows $d_{\parallel, ij}$ as a function of $\sigma_{ij}$ averaged over distances 8–15 µm. **f** Relative velocity of neighbors in a distance of 8–15 µm for different values of $\Gamma$ (color of the curve) for negative (left) and positive (right) $\Gamma$. Light colored areas are the standard deviation for different experimental runs. For negative $\Gamma$ we observe a repulsive interaction for high synchronization (positive relative velocity), while for positive $\Gamma$, synchronized swarmalators attract each other (negative relative velocity). In all experiments and simulations, 400 swarmalators were initialized in an area of size 400 × 280 µm, and allowed to relax until no two reference positions are within 10 µm separation using the exclusion algorithm in Methods section "Reference position update". This gave a density of around 0.003 µm⁻² at the end of the relaxation step. Source data are provided as a Source data file.

their dynamics within the general swarmalator framework. Specifically, we represent the phase coupling between oscillating colloids as a distance-dependent, Kuramoto-like coupling term (for more details, see SI section 4). With the oscillation frequency $\omega_0$ of isolated swarmalators, the corresponding equation of motion for the phase can be written as

$$\dot{\theta}_i(t) = \omega_0 + \sum_{j \neq i} \frac{K}{|\mathbf{q}_{ij}(t)|^3} \sin\left[\Delta\theta_{ij}(t)\right] + \xi_{i,\theta}(t). \tag{3}$$

Here, $\Delta\theta_{ij}(t)$ is the phase difference between neighboring swarmalators and $\xi_{i,\theta}(t)$ Gaussian, $\delta$-correlated noise. Opposed to frequently

used swarmalator models with long-range coupling[14], hydrodynamic interactions results in a shorter-ranged $|\mathbf{q}_{ij}|^{-3}$ decay of the synchronization[41].

As demonstrated in the previous section, the swarmalators in our experiments exhibit attraction or repulsion depending jointly on their relative position and phase synchronization. This interaction is mediated by the longitudinal displacement $d_{\parallel, ij}$ of each ABP along the direction of its neighbor. We find that $d_{\parallel, ij}$ can be well described by the empirical relation $d_{\parallel, ij} \propto |\mathbf{q}_{ij}|^{-2} \sin(\pi\sigma_{ij})$. With the assumption that the net displacement $\mathbf{d}_{i,t}$ is the superposition of displacements in the direction of neighbors, i.e., $\mathbf{d}_{i,t} = 1/N_j \sum_j d_{\parallel, ij} \cdot \hat{\mathbf{q}}_{ij}$, the motion of

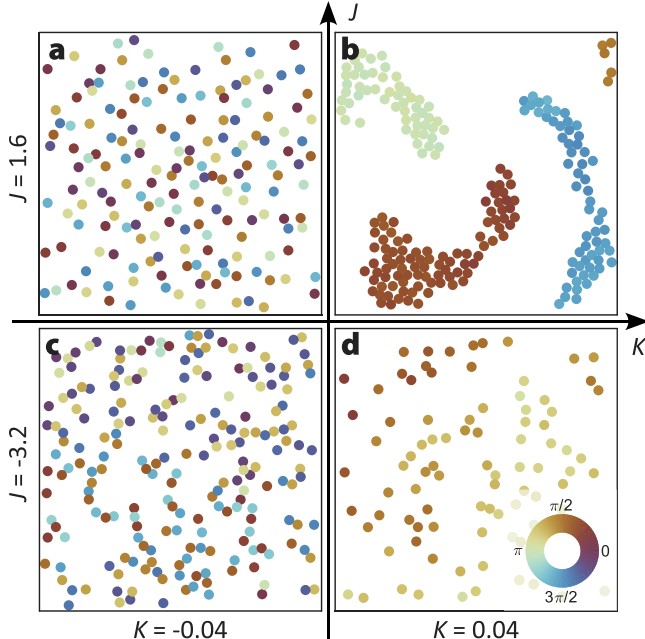

**Fig. 4 | Phase diagram of the swarmalator model.** $K$ and $J$ are the parameters governing the distance-dependent synchronization and synchronization-dependent attraction or repulsion, respectively (see equations (3) and (4)). **a** For $K < 0, J > 0$, the swarmalators arrange randomly without synchronization. **b** For $K > 0, J > 0$, highly synchronized clusters emerge. **c** For $K < 0, J < 0$, the swarmalators arrange in chain-like structures in which adjacent swarmalators have opposite phases. **d** For $K > 0, J < 0$, the swarmalators synchronize but repel each other. The colors denote the phases of the swarmalators as indicated in (**d**). For a quantitative description of the four phases, see SI Section 8. Additional snapshots for different parameter values are provided in Fig. S4, and an animated phase diagram is available in Supplementary video 3. For all $J, K$ values, the noise was fixed with an amplitude $D_\theta = D_\eta = 0.001$, where $\langle \xi_{i,\theta}(t) \xi_{j,\theta}(t') \rangle = D_\theta \delta(t - t')$ and $\langle \boldsymbol{\xi}_{i,\eta}(t) \cdot \boldsymbol{\xi}_{j,\eta}(t') \rangle = dD_\eta \delta(t - t')$, where $d$ is the dimensionality (here $d = 2$). We set $\omega_0 = 1$, integration time-step $dt = 0.005/\omega_0$, and ran the simulation for a total time of $20,000/\omega_0$.

swarmalators can be described as

$$\dot{\mathbf{q}}_i(t) = \frac{1}{N_j} \sum_{j \neq i} \frac{J \mathbf{q}_{ij}(t)}{|\mathbf{q}_{ij}(t)|^3} \sin\left[\pi \cos\left(\Delta \theta_{ij}(t)\right)\right] + \mathbf{F}_{rep,ij}(t) + \boldsymbol{\xi}_{i,q}(t). \tag{4}$$

$\boldsymbol{\xi}_{i,q}(t)$ is Gaussian, $\delta$-correlated noise, and $\mathbf{F}_{rep,ij}$ adds a repulsive force for small distances (see Methods section "Reference position update"). For more details on the motivation of equation (4), see SI section 7. By integrating equations (3) and (4) with $K > 0$ and varying $J$, we recover both the synchronized clusters and the dispersed states observed in experiments and simulations (Fig. 4; Fig. S4; Supplementary video 3). The model also enables exploration of anti-synchronizing interactions ($K < 0$), which are not accessible in our experiments due to the synchronizing interactions of the oscillators. Hydrodynamic anti-synchronization is, however, in principle possible and has been reported in other colloidal systems[34,42–44]. For $K < 0$ and $J > 0$, the system evolves into a dispersed, weakly synchronized state. In contrast, when both synchronization and spatial interaction are repulsive ($K < 0$, $J < 0$), swarmalators self-organize into chain-like aggregates in which adjacent particles oscillate in antiphase. We can rationalize these effectively one-dimensional structures considering that denser configurations lead to frustrated phase arrangements, destabilizing synchronization, and inducing repulsion between synchronized pairs. Similar alternating phases have also been observed in

numerical simulations of swarmalators with entirely different coupling and interaction mechanisms[45]. This suggests that the states observed in Fig. 4 are rather generic beyond the specific swarmalator realization in our study. By using synchronization, cluster size, and cluster asphericity as order parameters, the data for different $J$ and $K$ values separate into four distinct phases: synchronized clusters, dispersed-synchronized, chain-like anti-synchronized, and dispersed anti-synchronized states (see SI section 8).

## Outlook

Previous numerical studies of swarmalator systems have predominantly focused on quasi-static pattern formation, where steady-state configurations emerge rapidly. In contrast, our experimental system evolves on markedly longer timescales. Notably, the two principal observables—phase synchronization ($\sigma$) and the fraction of swarmalators in clusters ($f_c$)—exhibit different relaxation dynamics (Fig. 5a). Following a change in the coupling parameter $\Gamma$, synchronization decays rapidly (within a few oscillation periods $\tau_{osc}$), whereas structural reorganization, as measured by $f_c$, proceeds over tens of $\tau_{osc}$. This timescale separation gives rise to non-trivial state trajectories when $\Gamma$ is modulated dynamically. In particular, switching $\Gamma$ between positive and negative values generates a closed loop in the $(\sigma, f_c)$ state space (Fig. 5b), where the system transiently explores configurations that are inaccessible under static coupling. These intermediate states depend on the switching direction, reflecting a form of hysteresis. At higher switching rates, the system no longer equilibrates between transitions, but remains in a transient intermediate state, displaying a form of dynamic memory (Fig. 5c, d). Since this effect is a direct consequence of the asynchronous relaxation of spatial and phase degrees of freedom, we expect it to be a generic feature of swarmalators, which offers additional possibilities regarding the control of swarmalator systems[46].

Besides gaining control over swarmalator systems, another important direction in swarmalator research is the search for new types of interactions that can expand the range of phenomena these models can explain[47,48]. The swarmalators in our experiments reveal a qualitatively new interaction channel in the form of synchronization-dependent forces, which are not limited to the axis connecting swarmalators to which theoretical models are usually restricted. We find that the hydrodynamic interactions also give rise to lateral forces, i.e., forces acting orthogonal to the connecting axis, arising from an orthogonal component of particle displacement $d_\perp$ (Fig. 5e, f), depending on synchronization and distance. Similar to the parallel displacement, which gives rise to the synchronization-dependent attraction (see Fig. 3), this lateral displacement results in a rotational motion of swarmalators around each other. This kind of rotational motion originating from hydrodynamics is observed in multiple biological systems, like groups of starfish embryos[27], but has not been reproduced in swarmalator models before. In the phenomenological model in the section "Swarmalator model", we have omitted these lateral forces for simplicity, but this effect can be readily incorporated by adding another term to equation (4). Figure 5g, h shows a simulation snapshot of a rotating cluster of swarmalators and the size and $\Gamma$-dependent angular velocity. The rotation speed of the cluster can also be tuned by $\Gamma$, which couples the hydrodynamic interactions of the swarmalators to their motion. Importantly, the individual particles have no intrinsic chirality; instead, symmetry is broken through noise-assisted flips in rotation and synchronization mediated by hydrodynamic interactions, resulting in an emergent global rotation of the entire cluster with the same sense as its constituent swarmalators (see Fig. 5e, SI section 9). Incorporating lateral forces into swarmalator models fundamentally enriches the emergent phenomenology, enabling the formation of dynamic rotational states reminiscent of biological systems, such as vortical structures in bacterial colonies and starfish embryo aggregates[27,28]. Our results, therefore, expand the

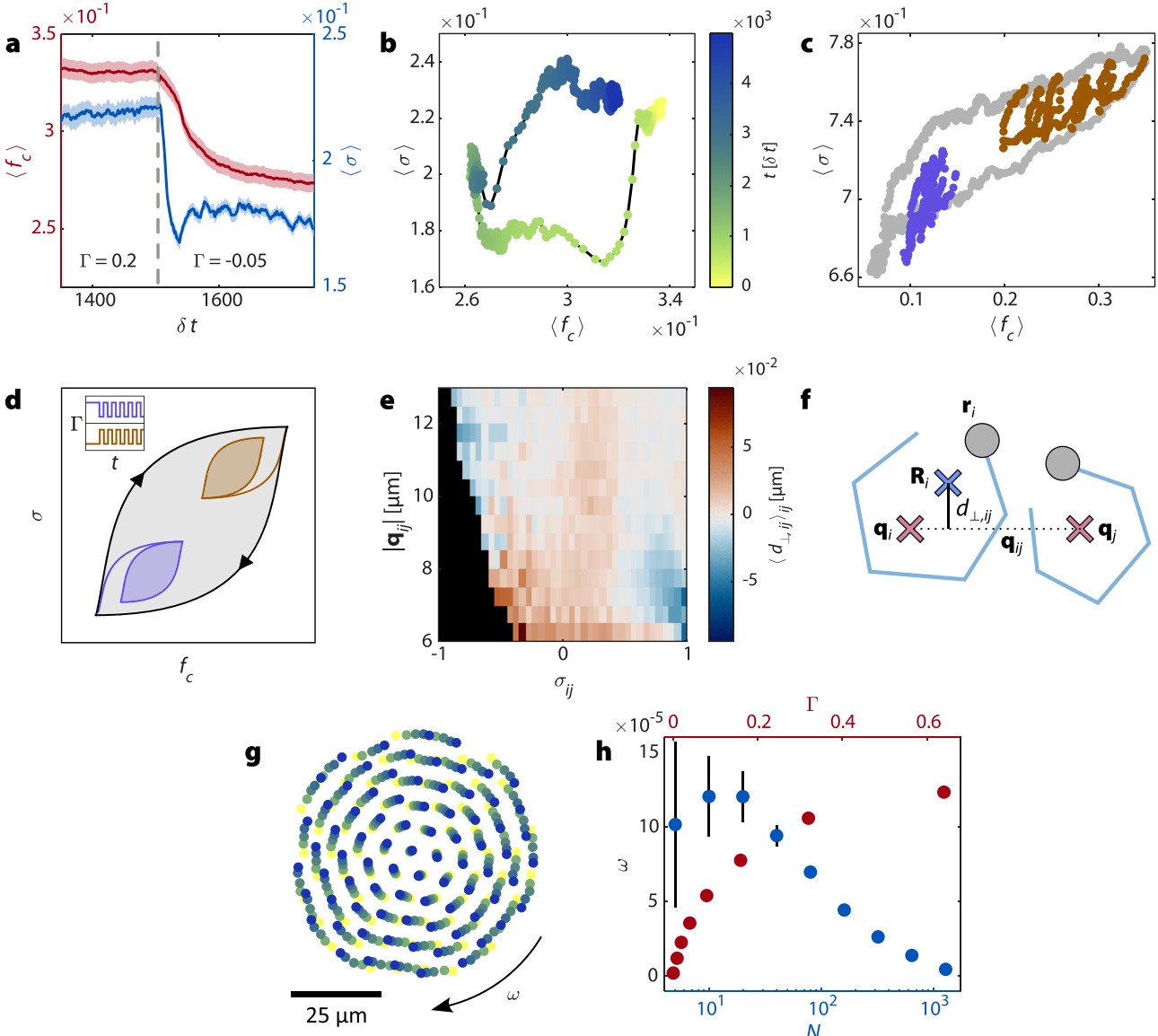

**Fig. 5 | Effects of time-varying interactions and lateral forces in swarmalators.** **a** The mean fraction of swarmalators in clusters $\langle f_c \rangle$ relaxes slower than the mean synchronization $\langle \sigma \rangle$ after a switch in $\Gamma$ (the light colored areas show the SEM over multiple runs). We selected $\Gamma = -05$ and $\Gamma = 0.2$ here, because these values result in the same swarmalator motility (see Fig. S11; SI section 6). **b** Switching back and forth between two values of $\Gamma$ results in a loop in the $\langle f_c \rangle - \langle \sigma \rangle$ space, because the system visits different states depending on the direction of the switching. $\langle \sigma \rangle$ and $\langle f_c \rangle$ are averaged over multiple runs. **c** State loop in the $\langle f_c \rangle - \langle \sigma \rangle$ space in simulations, for slow switching (every $400\delta t$) of $\Gamma$ (gray, as in (**b**)) as well as for fast switching (every $200\delta t$) from different initial $\Gamma$ (purple and orange). $\langle \sigma \rangle$ and $\langle f_c \rangle$ are averaged over 50 runs. **d** Illustration of the state loop with two smaller loops (purple and orange) that are a result of faster switching in $\Gamma$ from different initial values of $\Gamma$ (see inset). **e** Heat map of the averaged normal displacement $\langle d_{\perp, ij} \rangle_{ij}$ against the synchronization $\sigma_{ij}$ and reference point distance $|\mathbf{q}_{ij}|$ between neighboring swarmalators in experiment. **f** The normal (perpendicular) displacement $d_{\perp, ij}$ is defined as the component of the displacement $\mathbf{d}_i$ orthogonal to the direction to the neighboring reference point $\mathbf{q}_{ij}$, multiplied by the sign of the angular velocity of $i$. **g** Snapshots of a rotating swarmalator cluster in simulations at different times (colors yellow to green to blue). **h** Mean and standard deviation of the angular velocity $\omega$ of a rotating cluster as a function of the number $N$ of swarmalators in the system and the coupling parameter $\Gamma$ in simulations In (**g**, **h**), the swarmalators were initialized on a hexagonal grid with a separation of 8 μm, the system was allowed to relax for $3000\delta t$, and the data was collected and averaged over $30{,}000\delta t$. In (**a**, **b**, **c**), the initial density is around 0.003 μm$^{-2}$. Source data are provided as a Source data file.

physical scope of swarmalator dynamics and underscore the need to include anisotropic interaction channels in future theoretical frameworks.

## Methods

### Experimental realization of oscillating ABPs

The Janus colloids we used as ABPs were prepared by coating commercially available silica particles (radius 3 μm) with 80 nm of carbon on one hemisphere. We dispersed these particles in a binary mixture of water and 26.8 wt% 2,6-Lutidine, which has a critical demixing temperature of 34 °C. The dispersion was placed inside a quartz glass sample cell with a height of 200 μm, which was kept at 28 °C. Focusing a laser with a wavelength of 532 nm on these particles heats up the carbon cap, which leads to anisotropic demixing of the critical mixture around them and self-phoretic motion. The propulsion speed can be controlled via the laser intensity. We controlled the motion direction by offsetting the center of the laser spot from the particle center by 2.4 μm. Using an acousto-optical deflector, we scanned the laser beam over the particles at 100 kHz, allowing us to move multiple hundreds of ABPs in parallel. To achieve continuous steering of the particles, we captured microscope images, detected the particle positions, and steered each particle towards its assigned reference position in a

feedback loop at 2.5 Hz. As the particles move after the image capture at time $t$, the laser spots, which we apply at $t + \delta t$ ($\delta t = 0.4$ s), push them in directions misaligned with the reference position. This kind of time delay and direction mismatch in the steering signal leads to a continuous oscillating motion of the ABPs around their reference points[32]. The ABPs used in this study do, in principle, show negative phototaxis, i.e., their uncapped side and therefore their swimming direction reorient to point down light intensity gradients with reorientation times in the order of $1$ s[30]. However, we find that in our experiments, where we change the driving direction of each ABP with 2.5 Hz (i.e., a delay time of $\delta t = 0.4$ s), the colloids can not reorient sufficiently fast and are most of the time oriented with their carbon cap pointing in the direction orthogonal to the substrate they are moving on. As a result, the Janus colloids employed as ABPs here essentially behave like isotropic active colloids.

The relaxation time of the propulsion mechanism of the ABPs is around $0.1$ s[30]. Since the time $\delta t$ in between changes of the laser spot positions is larger than this propulsion relaxation time, the ABPs react quasi-instantaneously to changes in the driving direction.

To compensate for the mechanical instabilities of the optical setup that result in slight misalignments between the laser focus and the ABP positions, we carried out periodical recalibrations throughout all our measurements.

## Simulation model

The motion of the swarmalators is described by an ABP model of self-propelled particles with hydrodynamic interactions. An ABP experiences a propulsion force $f_p$ acting along an orientation vector $\mathbf{e}_i$. The motion of such an ABP at position $\mathbf{r}_i$ is governed by

$$m\ddot{\mathbf{r}}_i = f_p \mathbf{e}_i - \gamma \dot{\mathbf{r}}_i + \sqrt{2D_t\gamma^2}\boldsymbol{\zeta}_i + \mathbf{F}_{HI}(\mathbf{r}_i), \quad (5)$$

where $\gamma$ is the damping coefficient related to the fluid viscosity, $D_t$ is the thermal translational diffusion coefficient of the particle, and $\boldsymbol{\zeta}_i$ is a Gaussian random process with $\langle \boldsymbol{\zeta}_i(t) \rangle = 0$ and $\langle \boldsymbol{\zeta}_i(t)\boldsymbol{\zeta}_j(t') \rangle = \delta_{ij}\delta(t - t')$. For a particle speed of $u_0$, the required propulsion force is given by $f_p = \gamma u_0$. The swarmalators interact hydrodynamically via the interaction $\mathbf{F}_{HI}(\mathbf{r}_i) = \gamma \mathbf{U}_{HI}(\mathbf{r}_i)$, due to the movement of the fluid by each swarmalator. Here, the flow field $\mathbf{U}_{HI}(\mathbf{r}_i)$ is given by

$$\mathbf{U}_{HI}(\mathbf{r}_i) = \sum_j \mathbf{U}_{flow,j}(\mathbf{r}_i - \mathbf{r}_j) + \mathbf{U}_{lub,j}(\mathbf{r}_i - \mathbf{r}_j) \quad (6)$$

where the sum is over all particles $j$ within some cutoff distance $R_{HI}$, i.e., $|\mathbf{r}_i - \mathbf{r}_j| < R_{HI}$. The first term $\mathbf{U}_{flow,j}$ is the hydrodynamic force generated at the position of particle $i$ due to the self-propulsion of the particle $j$, and is modeled using the squirmer flow field[49,50]

$$\mathbf{U}_{flow,j}(r) = -\frac{p}{r^2}[1 - 3(\mathbf{e}_j \cdot \hat{\mathbf{r}})^2]\hat{\mathbf{r}} - \frac{s}{r^3}[\mathbf{e}_j - 3(\mathbf{e}_j \cdot \hat{\mathbf{r}})\hat{\mathbf{r}}] \quad (7)$$

with $\mathbf{r} = \mathbf{r}_i - \mathbf{r}_j$, $r = |\mathbf{r}|$ and $\hat{\mathbf{r}} = \mathbf{r}/r$. The first term decaying as $1/r^2$ is due to a force dipole of strength $p = -3\beta u_0 a^2/4$, and the second term corresponds to a source dipole of strength $s = u_0 a^3/2$, where $\beta$ is the squirmer parameter that measures the asymmetry in the flow field and $a$ is the particle radius. For $\beta > 0$, we have a puller, and for $\beta < 0$, we have a pusher. For light-activated colloids, we expect no strong dipole contributions, i.e., $|\beta| < 1$.

In addition to the squirmer (far-field) interactions, we also include (near-field) lubrication forces between swarmalators[51], as squirmer interactions alone are insufficient to account for the strong hydrodynamic interactions observed in the experiment (see SI section 2;

**Table 1 | Table showing the simulation parameters and the corresponding value in the experiment**

| Parameter | Simulation units | Experimental value |
| --- | --- | --- |
| Particle radius $a$ | 3 | 3.0 μm |
| Diffusion coefficient $D_t$ | 0.03 | 0.03 μm s⁻² |
| Particle velocity $u_0$ | 2.0 | 2 μm s⁻¹ |
| Delay time $\delta t$ | 0.4 | 0.4 s |
| Reference point exclusion distance $R_{max}$ | 10 | 10 μm |
| Laser distance $l_0$ | 3 | 2.0–3.0 μm |
| Hydrodynamic cutoff $R_{HI}$ | 30 | - |
| Lubrication length scale $R_0$ | 3.5$a$ | - |
| Relaxation time $m/\gamma$ | 0.01 | - |
| Integration time $dt$ | 0.0025 | - |

Lengths in the simulation are given in units of 1 μm, and time in units of 1 s.

Fig. S6). The flow field related to lubrication forces is given by

$$\mathbf{U}_{lub,j}(r) = -\frac{a}{h}(\dot{\mathbf{r}} \cdot \hat{\mathbf{r}})\Theta(r)\hat{\mathbf{r}} \quad (8)$$

where $\dot{\mathbf{r}} = \dot{\mathbf{r}}_i - \dot{\mathbf{r}}_j$ is the relative velocity between particle $j$ and $i$, $h = r - 2a$ is the shortest distance between the surface of the two particles, and $\Theta(r) = \exp(-r/R_0)$ is a cutoff function whose decay radius $R_0$ is determined by matching the data to the experiment (see SI section 2; Fig. S6).

The ABP orientation vector $\mathbf{e}_i(t)$ at any time instant $t$ is determined by the position of the laser spot $\mathbf{l}_i(t)$, so that $\mathbf{e}_i(t) = (\mathbf{r}_i(t) - \mathbf{l}_i(t))/|\mathbf{r}_i(t) - \mathbf{l}_i(t)|$, as the particle moves in the direction away from the laser spot. By choosing this update scheme that matches the experimental setup, we are able to closely replicate the experimental conditions and obtain very good agreement for the single particle motion (see SI section 1; Fig. S5). The laser spot is updated every delay time $\delta t$, with the update rule

$$\mathbf{l}_{i,t+\delta t} = \mathbf{r}_{i,t+\delta t} - l_0 \frac{(\mathbf{q}_{i,t} - \mathbf{r}_{i,t})}{|\mathbf{q}_{i,t} - \mathbf{r}_{i,t}|} \quad (9)$$

where $l_0$ is the distance from the particle center at which the laser is illuminated, and $\mathbf{q}_{i,t} - \mathbf{r}_{i,t}$ is the displacement vector between the particle's reference position $\mathbf{q}_i$ and the particle's position $\mathbf{r}_i$ one delay time prior, i.e., at $(t + \delta t) - \delta t$. The system lies in the regime of low Reynolds number, so that we simulate equation (5) in the over-damped regime, i.e., $m/\gamma \ll \delta t$, so that effectively

$$\dot{\mathbf{r}}_i \simeq v_0 \mathbf{e}_i + \sqrt{2D_t}\boldsymbol{\zeta}_i + \mathbf{U}_{HI} \quad (10)$$

The equation of motion (5) is solved using the Velocity-Verlet integration scheme[52]. All the simulation parameters along with the corresponding experimental values are collected in Table 1.

## Reference position update

To couple the motion of the swarmalator to its rotational oscillations, a reference-point update rule based on the swarmalators past trajectory is implemented, with reference position $\mathbf{q}_{i,t}$ updated every delay time as

$$\mathbf{q}_{i,t+\delta t} = \mathbf{q}_{i,t} + \Gamma \cdot (\mathbf{R}_{i,t} - \mathbf{q}_{i,t}) \quad (11)$$

where $\mathbf{R}_{i,t} = \frac{1}{T}\sum_{t-T}^{t}\mathbf{r}_{i,t}$ is the average position of the swarmalator measured over the last $T$ time steps, and $\Gamma$ controls the strength of the displacement of the new reference position. Moreover, to prevent the reference positions from coming too close together—which could cause collision between particles—an exclusion algorithm between the

reference positions is implemented following a Monte-Carlo procedure. Thus, every reference update $\mathbf{q}_{i,t}$ is accepted with probability

$$P_i = \exp(-\Delta E / k_b T) \tag{12}$$

where

$$\Delta E = \sum_{i \neq j} V_{\mathrm{LJ}}(|\mathbf{q}_{i,t} - \mathbf{q}_{i,t-\delta t}|) - V_{\mathrm{LJ}}(|\mathbf{q}_{i,t-\delta t} - \mathbf{q}_{i,t-\delta t}|), \tag{13}$$

and $V_{\mathrm{LJ}}$ is the truncated Lennard-Jones potential given by

$$V_{\mathrm{LJ}}(r) = \begin{cases} 4\varepsilon \left( \dfrac{R_{\max}}{r} \right)^{12} & r \leq R_{\max} \\ 0, & r > R_{\max} \end{cases} \tag{14}$$

Here $R_{\max}$ is the minimum desired reference-point separation. The particle distribution around the reference points indicates that particles can reach distances of up to 2.0 μm from their reference positions (see Fig. S5a), thereby suggesting a minimum center-to-center separation of $R_{\max} = 10$ μm (particle radius 3 μm) for the reference points to avoid overlapping ABPs. We therefore set $\epsilon/k_b T = 0.2$, so that while particles typically maintain a separation of $R_{\max}$, strong hydrodynamic forces can bring the particles closer together. Given the initialization density is around 0.003 μm$^{-2}$, for weak or no clustering ($\Gamma \simeq 0.0$), the clustering coefficient is expected to remain low, as there is, on average, only one particle per box of side length 18 μm. Therefore, we choose a cutoff of 15 μm to measure clustering in the system. Additionally, in simulations, particles closer than 6.25 μm are excluded from the analysis, since lubrication forces become too strong and the behavior is uncontrolled and ill-defined.

## Data availability
The experiment and simulation data that support the results of this study is available at Zenodo DOI:10.5281/zenodo.15862310[53]. Source data are provided with this paper.

## Code availability
The simulation code and analysis scripts that were used in this study are available at Zenodo DOI:10.5281/zenodo.15862310[53].

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

## Acknowledgements
V.L.H. and C.B. acknowledge funding from the DFG Centre of Excellence 2117, Germany Centre for the Advanced Study of Collective Behavior (ID:422037984), and C.B. acknowledges funding by the ERC through the Adv. Grant BRONEB (101141477).

## Author contributions
V.L.H. and P.I. contributed equally to this work. V.L.H. and C.B. conceptualized the study. V.L.H. performed experiments and data analysis, and prepared the figures. P.I. and G.G. designed the simulations. P.I. performed numerical simulation and data analysis. All authors contributed to the discussion of the results and writing of the paper.

## Funding

## Competing interests
The authors declare no competing interests.
