## [Transparent Peer Review file · Nature Communications]

Tunable colloidal swarmalators with hydrodynamic coupling

Corresponding Author: Professor Clemens Bechinger

Version 0:

Reviewer comments:

Reviewer #1

(Remarks to the Author)

The authors studied experimental realization of swarmalators using feedback-controlled self-propelling colloidal particles. By tuning a single parameter, they observed the co-evolution of spatial and phase dynamics, leading to a variety of collective states such as synchronized clusters, rotating aggregates, and dispersive phases. Importantly, they demonstrated that rapid changes of this parameter can generate dynamic regimes inaccessible to systems with static interactions. Simulations with squirmer and lubrication forces supported the experimental results, and a new synchronization-dependent interaction channel was also identified.

The manuscript sounds incomplete except the following improvements:

1. The study utilizes active Brownian particles (ABPs), realized using silica spheres suspended in a near-critical water–lutidine mixture. However, the rationale for selecting this particular system is not clearly discussed in the manuscript and should be elaborated.
2. The authors stated that operating several such oscillators with small distances $q_{\{ij\}}$ between their reference points, their oscillation becomes highly synchronized (Fig. 1d). However, it is not immediately clear from Fig. 1d how this conclusion can be drawn. The authors should provide a brief explanation to clarify this point.
3. As a measure of synchronization σ_j authors defined in equation (1). But N_j is not defined anywhere in the manuscript. The authors should clarify.
4. The authors presented the mean squared displacement (MSD) of isolated oscillators and its plot in Fig. 2(c). However, the definition of MSD and the justification for its use in the analysis are not clearly provided and should be clarified.
5. In Fig. 3c, the fraction of oscillators within clusters is shown to increase continuously with Γ . However, in the region $0 < \Gamma < 0.10$ the figure indicates a decreasing trend in the fraction of clustered oscillators. The authors should highlight and discuss this feature. The authors should also clarify whether the clustering behavior depends on the distance between neighboring oscillators.
6. To clearly demonstrate the dependence of $d_{\{ij\}}$ on both distance and synchronization it would be more effective if authors include a plot of $d_{\{ij\}}$ vs $\sigma_{\{ij\}}$ in the manuscript.
7. In the swarmalator model [Eqs. (3) and (4)], authors observed emerging states for different parameter values, It would strengthen the work if a parameter space varying K and J is included, showing the regions where different collective states appear.
8. Mention the initial conditions and methods for simulations.
9. In the caption of Fig. 2, the word ‘of’ appears twice and should be corrected. In addition, the figure captions throughout the manuscript could be enhanced to provide clearer explanations of the figures. The quality of the figures themselves may also be improved for better clarity.
10. I would like to note recent reviews that would fit well in the introduction section on swarmalators <https://doi.org/10.1098/rspa.2025.0315> and SIAM Journal on Applied Mathematics 85 (4), 1475-1499 (2025).

(Remarks on code availability)

The authors studied experimental realization of swarmalators using feedback-controlled self-propelling colloidal particles. By tuning a single parameter, they observed the co-evolution of spatial and phase dynamics, leading to a variety of collective states such as synchronized clusters, rotating aggregates, and dispersive phases. Importantly, they demonstrated that rapid changes of this parameter can generate dynamic regimes inaccessible to systems with static interactions. Simulations with squirmer and lubrication forces supported the experimental results, and a new synchronization-dependent

interaction channel was also identified.

The manuscript sounds incomplete except the following improvements:

1. The study utilizes active Brownian particles (ABPs), realized using silica spheres suspended in a near-critical water–lutidine mixture. However, the rationale for selecting this particular system is not clearly discussed in the manuscript and should be elaborated.
2. The authors stated that operating several such oscillators with small distances q_{ij} between their reference points, their oscillation becomes highly synchronized (Fig. 1d). However, it is not immediately clear from Fig. 1d how this conclusion can be drawn. The authors should provide a brief explanation to clarify this point.
3. As a measure of synchronization σ_j authors defined in equation (1). But N_j is not defined anywhere in the manuscript. The authors should clarify.
4. The authors presented the mean squared displacement (MSD) of isolated oscillators and its plot in Fig. 2(c). However, the definition of MSD and the justification for its use in the analysis are not clearly provided and should be clarified.
5. In Fig. 3c, the fraction of oscillators within clusters is shown to increase continuously with Γ . However, in the region $0 < \Gamma < 0.10$ the figure indicates a decreasing trend in the fraction of clustered oscillators. The authors should highlight and discuss this feature. The authors should also clarify whether the clustering behavior depends on the distance between neighboring oscillators.
6. To clearly demonstrate the dependence of $d_{\{ij\}}$ on both distance and synchronization it would be more effective if authors include a plot of $d_{\{ij\}}$ vs σ_{ij} in the manuscript.
7. In the swarmalator model [Eqs. (3) and (4)], authors observed emerging states for different parameter values, It would strengthen the work if a parameter space varying K and J is included, showing the regions where different collective states appear.
8. Mention the initial conditions and methods for simulations.
9. In the caption of Fig. 2, the word ‘of’ appears twice and should be corrected. In addition, the figure captions throughout the manuscript could be enhanced to provide clearer explanations of the figures. The quality of the figures themselves may also be improved for better clarity.
10. I would like to note recent reviews that would fit well in the introduction section on swarmalators <https://doi.org/10.1098/rspa.2025.0315> and SIAM Journal on Applied Mathematics 85 (4), 1475-1499 (2025).

Reviewer #2

(Remarks to the Author)

In the submitted manuscript, the authors presents a controllable experimental swarmalator system using feedback-controlled self-propelled colloidal particles coupled by hydrodynamic flows. Combining experimental observation, simulation and theory, they show that the synchronization and spatial dynamics in the system together give rise to several collective states including synchronized clusters, rotating aggregates, and dispersive phases controlled by a coupling parameter. A new interaction synchronized from the direction perpendicular to the connection axis of particles are also revealed. Overall, the results are interesting and helpful for the understanding the swarmalator systems. However, there are a few comments for the authors to improve the manuscript.

1. There is a big issue about the output of the figures in the main text. All color bars in figures (Fig. 1f, Fig. 3b, Fig. 3e, Fig. 5b, Fig. 5f, Fig. 6c) and heat maps (Fig. 3e, Fig. 5f) are mistakenly shifted somehow. They are confusing. However, those in the SI and videos seem all right.
2. The particles used are Janus particles with one hemisphere covered by the carbon. While some reference work used uniform gold nanoparticles. Does the distribution of the heat-absorbing materials affect the particle dynamics? Will the orientation of carbon caps be considered and detected for each time interval?
3. What is the relaxation time of the flow generated by the heating and particle movement? The time delay is 0.4 s. What happens if the time delay is tuned, especially around the flow relaxation time?
4. The density or the average distance in each realization needs to be clarified since it is also a controlled parameter. The definition of clusters uses the criterium of 15 microns. Particles are not quite close considering the radius of the particles is 3 microns. Any reason for choosing this number?
5. It seems that there is a small oscillation flow between left bottom to right up initially in the experiment part of Video 2. Is it true (and why?) or just a visual misperception?
6. There is not experimental counterpart of the mini state loops shown as shown in Fig. 5d of the simulation results. How is the experimental one?

(Remarks on code availability)

Reviewer #3

(Remarks to the Author)

The manuscript presents experimental work on the collective behavior of active Brownian particles that simultaneously exhibit synchronized rotation and translational motion in a coupled fashion, corroborated with numerical simulations and theoretical modelling. A several hundred colloidal particles perform orbiting motion around the reference position of each, steered by a scanning laser with a feedback loop. The reference positions are also moved to induce translational motion of

the particles controlled by a coupling parameter (Γ); the reference position of each particle decays toward its average position in the short past for $\Gamma > 0$, while it is moved away from the average position for $\Gamma < 0$.

When the reference positions are fixed on a hexagonal lattice, the particles exhibit synchronized rotation with the average degree of synchronization negatively correlated with the lattice spacing. For motile reference positions, in-phase synchronized pairs tend to attract each other, while anti-phase pairs repel each other. Simulations based on the squirmer model with short-range lubrication interactions reproduce the experimental observation qualitatively, if the swimmers are of the puller-type. For pusher-type swimmers, the correlation between the displacement and degree of synchronization behaves differently.

The authors also introduce a phenomenological swarmalator model with a long-range Kuramoto-type phase coupling and forces that depend on the phase difference between particles. The coupling parameters K for the phase and J for the repulsion/attraction control the collective phase behavior, which is qualitatively consistent with the experimental results for positive (synchronizing) K .

Furthermore, (i) time-varying interactions are realized in the experiment, and a fast switching between positive and negative Γ caused bistable collective states; (ii) lateral forces that are perpendicular to the axis connecting two particles causes global rotation of a cluster in the simulations.

The manuscript reports on a state-of-the-art experiment that provides the first controllable realization of a swarmalator system. The results are highly interesting and represent a significant and novel contribution to the emerging field at the intersection of nonlinear science and active matter physics. The qualitative agreement between the experimental results and the numerical simulations is impressive given the system's complexity. Therefore, I can recommend this manuscript for publication in Nature Communications provided the authors can satisfactorily address the following points.

Major issues:

1. The derivation and justification of the phenomenological swarmalator model [Eqs. (3,4)] should be clarified. The authors should provide a more detailed explanation, perhaps in the Supplementary Material, that addresses the following points:

(i) The $1/r^3$ coupling should be associated with the no-slip boundary condition at the substrate, and the coupling parameter K should be related to the distance from the substrate and other parameters.

(ii) There appears to be a contradiction between the squirmer model, which is based on the Oseen tensor for bulk fluid, and the $1/r^3$ coupling in the swarmalator model, which is associated with hydrodynamic interactions near a no-slip boundary. The authors should elaborate on the connection between these two descriptions or justify the use of the $1/r^3$ form in their phenomenological model.

(iii) The approximation for the longitudinal distance d_{\parallel} needs justification by experimental data. It is unclear how this quantity is related to the cosine term in Eq.(4).

2. The phase diagram presented in Fig. 4 is currently depicted as a collection of representative snapshots. To make this a quantitative and more rigorous phase diagram, the authors should define and apply quantitative order parameters to clearly delineate the boundaries between the different collective states (e.g., static synchronous, rotating cluster, async).

3. The emergence of global cluster rotation from lateral forces [Fig. 5g,h] is an intriguing result that requires further explanation. The authors should clarify: (i) What parameters control the magnitude and speed of this global rotation? (ii) How is the direction of the collective rotation (e.g., clockwise or counter-clockwise) determined by the properties or chirality of the individual particles.

Minor issues:

- To place their findings in a broader context, the authors might consider discussing the observed global cluster rotation in relation to similar phenomena in biological systems, such as the collective rotation of starfish embryos [26].

- Optical feedback control to steer rotation of colloidal particles and switch synchronization patterns was studied in: A. Maestro et al, Commun. Physics 1, 28 (2018). The paper seems more relevant to the present work than the other quoted examples of anti-phase hydrodynamic synchronization.

- In page 5, left column, "Figure 3d presents the average longitudinal displacement ...", it should refer to Fig. 3e.

- The emergence of global cluster rotation from lateral forces [Fig. 5g,h] is an intriguing result that requires further explanation. The authors should clarify: (i) What parameters control the speed of rotation? (ii) How is the direction of the collective rotation (e.g., clockwise or counter-clockwise) related to the rotating direction and other properties of the individual particles.

- In Fig.S5, caption, "the data is shows" might be a typo.

(Remarks on code availability)

Version 1:

Reviewer comments:

Reviewer #1

(Remarks to the Author)

Authors have revised the manuscript nicely.

Now it can be accepted for publication.

(Remarks on code availability)

The results are seems correct from my best knowledge. The codes I didn't check.

Reviewer #2

(Remarks to the Author)

I am good with the response and recommend the draft for a publication.

(Remarks on code availability)

Reviewer #3

(Remarks to the Author)

The revised manuscript presents a significantly improved and more comprehensive study of tunable colloidal swarmalators with hydrodynamic coupling. The authors have clarified the physical basis of the phenomenological swarmalator model, linking the $1/r^3$ phase coupling to hydrodynamic interactions near a no-slip boundary and providing experimental validation for the longitudinal displacement relation. They have also introduced quantitative order parameters — cluster size, asphericity, and synchronization — to define the phase diagram, resolving the earlier concern that the classification was only qualitative. The explanation of global rotation and lateral forces has been expanded and now includes clear dependence on the coupling parameter Γ and cluster size. The simulation codes look correct and consistent with the models described in the manuscript.

Overall, the revision successfully addresses all of my previous comments. The experimental realization, numerical simulation, and theoretical modeling are now coherently integrated, and the results are presented with improved clarity and rigor. The manuscript now represents a definitive and publishable demonstration of a controllable swarmalator system. I therefore recommend its publication in Nature Communications.

(Remarks on code availability)

Response to the referee reports for the manuscript
NCOMMS-25-57131-T *Tunable colloidal swarmalators with hydrodynamic coupling*

Reviewer 1

The authors studied experimental realization of swarmalators using feedback-controlled self-propelling colloidal particles. By tuning a single parameter, they observed the co-evolution of spatial and phase dynamics, leading to a variety of collective states such as synchronized clusters, rotating aggregates, and dispersive phases. Importantly, they demonstrated that rapid changes of this parameter can generate dynamic regimes inaccessible to systems with static interactions. Simulations with squirmer and lubrication forces supported the experimental results, and a new synchronization-dependent interaction channel was also identified.

R1.1 The study utilizes active Brownian particles (ABPs), realized using silica spheres suspended in a near-critical water–lutidine mixture. However, the rationale for selecting this particular system is not clearly discussed in the manuscript and should be elaborated.

Reply The investigation of a real-world system of swarmalators with physical interactions requires the control of the motion of hundreds of moving agents which must physically interact with each other. Systems of self-propelled objects have been suggested as ideal candidates because they exactly fulfill such requirements and have shown basic swarmalator characteristics before (see e.g. Yan et al., Nature 2012, 491, 578-581 and Zhang et al., Nat Commun 2020, 11, 4401). Compared to other active moving agents (e.g. macroscopic robotic devices or drone swarms), active colloidal particles can be experimentally realized in table-top experiments under controlled lab conditions. A particular advantage of the system of light-activated active colloids, which is used in our study, is that the particle motion can be precisely controlled by a scanning laser beam, offering precise and individual control over the motion of the individual agents. In addition, due to the absence of a chemical fuel (the self-propulsion mechanism is created by light), this allows for experiments over several days – crucial to obtain sufficient statistics which makes this system very convenient for swarmalator studies. We have added an explanation regarding our choice of this experimental platform in section 2 of the manuscript.

R1.2 The authors stated that operating several such oscillators with small distances q_{ij} between their reference points, their oscillation becomes highly synchronized (Fig. 1d). However, it is not immediately clear from Fig. 1d how this conclusion can be drawn. The authors should provide a brief explanation to clarify this point.

Reply Yes, we agree that the image alone does not immediately convey this point. The conclusion can, however, be drawn quantitatively from the parameter σ_j . We have revised the text to clarify this point and added a reference to an Supplementary video 1, in which the synchronization can be perceived more clearly.

R1.3 As a measure of synchronization σ_j authors defined in equation (1). But N_j is not defined anywhere in the manuscript. The authors should clarify.

Reply We have now added a line defining N_j in the manuscript.

R1.4 The authors presented the mean squared displacement (MSD) of isolated oscillators and its plot in Fig. 2(c). However, the definition of MSD and the justification for its use in the analysis are not clearly provided and should be clarified.

Reply The MSD quantifies the dynamical regime of motion (diffusive, subdiffusive, superdiffusive/ballistic, confined) and is widely used in soft matter, biophysics, and statistical physics to classify particle dynamics. We agree that a proper mathematical definition is required which has now be added to the SI. In particular in the context of swarmalators, the MSD is frequently used to characterize the swarmalators activity and in particular its dependence on the coupling parameter Γ . To facilitate the comparison of our findings with other swarmalator studies, we have also used this measure in our work.

R1.5 In Fig. 3c, the fraction of oscillators within clusters is shown to increase continuously with Γ . However, in the region $0 < \Gamma < 0.10$ the figure indicates a decreasing trend in the fraction of clustered oscillators. The authors should highlight and discuss this feature. The authors should also clarify whether the clustering behavior depends on the distance between neighboring oscillators.

Reply The small dip in the cluster fraction f_c noted by the referee originates from our initial exclusion of pairs separated by less than $7 \mu\text{m}$ in the cluster analysis. This cutoff was introduced to exclude near-contact pairs ($r \approx 2a$) which generate very large lubrication forces in simulations. However, following the comment of the referee, we have reexamined the data and found that only pairs closer than about $6 \mu\text{m}$ actually exhibit such effects, while the $7 \mu\text{m}$ threshold unnecessarily excluded physically valid pairs. At small Γ , where motility is low, this resulted in an artificial reduction of f_c as particles tend to cluster near $7 \mu\text{m}$. Thus, the dip is actually an artifact resulting from an improper choice of the cutoff distance. We have therefore reduced the cutoff (for exclusion in the clustering analysis) to $6.25 \mu\text{m}$, which removes only near-contact pairs and avoids spurious exclusions. With this correction, f_c now increases smoothly and continuously in this region, as expected. We have added a few lines in the method section to clarify this calculation.

The synchronization-dependent attraction that leads to clustering in our system does indeed depend on the distance between swarmalators. The heat-map in Fig. 3e displays the exact dependence of the clustering behaviour on both the neighbour distance $|q_{ij}|$ and on their synchronization σ_{ij} , and Γ couples to the displacement d_{ij} to induce clustering in the system.

R1.6 To clearly demonstrate the dependence of $d_{\parallel,ij}$ on both distance and synchronization it would be more effective if authors include a plot of $d_{\parallel,ij}$ vs σ_{ij} in the manuscript.

Reply The main purpose of Fig. 3e was to highlight that the attractive and repulsive interactions that emerge from the hydrodynamics between ABPs depend on both the distance and the synchronization between the swarmalators, as usually assumed in swarmalator models. However, we agree that a heat map is not the most effective way to show the dependency of $d_{\parallel,ij}$ on σ_{ij} and have therefore added a plot of both the parallel and normal displacement against the synchronization averaged over the same distances as for the relative velocities in Fig. 3f ($8\text{-}15 \mu\text{m}$) in the new Extended Data Fig. 7. In this form, the dependency of the displacement on the synchronization can be more easily perceived.

R1.7 In the swarmalator model [Eqs. (3) and (4)], authors observed emerging states for different parameter values, It would strengthen the work if a parameter space varying K and J is included, showing the regions where different collective states appear.

Reply We thank the referee for this suggestion. In the Supplementary Information, we had previously included only a qualitative extension of the phase space. In response, we have now constructed an order-parameter space – defined by synchronization, cluster size, and cluster asphericity – that allows us to identify and distinguish the various swarmalator phases. Our analysis shows that the data separate clearly into four phases, corresponding to the representative states presented earlier. Using these order parameters, we are now able to present the swarmalator phases in the form of a phase diagram. We have included this analysis in the Supplementary Information, as it is somewhat technical and does not add significantly beyond the qualitative presentation in Fig. 4. The four phases occupy the four quadrants of the JK parameter space (corresponding to the positive and negative values of J and K), which is captured qualitatively by the phase diagram in Fig. 4.

R1.8 Mention the initial conditions and methods for simulations.

Reply The Methods section already provides some description of the simulation model and parameters (Section 7.2,7.3, Table 1). We have further added some details on initialization for the different cases and data acquisition in the figure captions and Supplementary Information.

R1.9 In the caption of Fig. 2, the word ‘of’ appears twice and should be corrected. In addition, the figure captions throughout the manuscript could be enhanced to provide clearer explanations of the figures. The quality of the figures themselves may also be improved for better clarity.

Reply We have corrected the typo. In addition, we have added further information to the figure captions. We agree that the figures were quite dense and have increased their size and added explanatory schematics for better clarity.

R1.10 I would like to note recent reviews that would fit well in the introduction section on swarmalators <https://doi.org/10.1098/rspa.2025.0315> and SIAM Journal on Applied Mathematics 85 (4), 1475-1499 (2025).

Reply Many thanks for pointing out these articles about non-Kuramoto and higher-order interactions between swarmalator. We have added them to outlook section of our manuscript.

Reviewer 2

In the submitted manuscript, the authors presents a controllable experimental swarmalator system using feedback-controlled self-propelled colloidal particles coupled by hydrodynamic flows. Combining experimental observation, simulation and theory, they show that the synchronization and spatial dynamics in the system together give rise to several collective states including synchronized clusters, rotating aggregates, and dispersive phases controlled by a coupling parameter. A new interaction synchronized from the direction perpendicular to the connection axis of particles are also revealed. Overall, the results are interesting and helpful for the understanding the swarmalator systems.

Reply We are very pleased by the referee's positive assessment of our manuscript.

R2.1 There is a big issue about the output of the figures in the main text. All color bars in figures (Fig. 1f, Fig. 3b, Fig. 3e, Fig. 5b, Fig. 5f, Fig. 6c) and heat maps (Fig. 3e, Fig. 5f) are mistakenly shifted somehow. They are confusing. However, those in the SI and videos seem all right.

Reply During submission of the manuscript we did not encounter any issues. We have, however realized now that on Mac devices, most heat maps and color bars are distorted and think the problem was the format in which we exported these graphics from Matlab. We are very sorry for this issue and have tried to adjust the figures of the revised version to solve this issue.

R2.2 The particles used are Janus particles with one hemisphere covered by the carbon. While some reference work used uniform gold nanoparticles. Does the distribution of the heat-absorbing materials affect the particle dynamics? Will the orientation of carbon caps be considered and detected for each time interval?

Reply The Janus colloids we employ as ABPs in our study do in principle align their carbon cap towards the laser, resulting in negative phototaxis, and therefore show different propulsion dynamics compared to isotropic active particles like, e.g., colloids evenly coated with gold nanoparticles. In our experiments, we do, however, change the relative position of the laser spot with respect to each particle at a higher frequency than the Janus particles can follow given their finite reorientation time (around 1 s, see Lozano et al., Nat Commun 2016, 7, 12828). As a result, the particles are always found with their carbon cap oriented vertically (i.e. parallel to the laser beam) and their bright and dark hemispheres can not be distinguished. For this reason, the Janus particles behave like isotropic active colloids in the oscillating mode in which they are driven in this study. To reduce the image processing time and because the particle orientation did not play an important role for the dynamics, we did not track the orientations of the particles in our experiments. We have added an additional paragraph for clarification in the Methods section.

R2.3 What is the relaxation time of the flow generated by the heating and particle movement? The time delay is 0.4 s. What happens if the time delay is tuned, especially around the flow relaxation time?

Reply The relaxation time of the propulsion mechanism is about 0.1 s (see Lozano et al., Nat Commun 2016, 7, 12828). Tuning the delay time between the detection of the particle positions and the propulsion towards the reference positions changes the oscillation frequency, the amplitude and the persistence of the orbiting motion of the ABPs (with a small delay time, the ABPs switch their direction of rotation less often). For our experiments, we have optimized the delay time to achieve oscillations with high amplitude and long correlation times of the angular velocity (i.e. few switches in rotation direction). The time it takes to process the microscope image and detect the particle positions sets a lower limit for the delay time we can achieve of 0.3 s. Thus, we were unfortunately not able to test delay times below the hydrodynamic relaxation time of the propulsion mechanism. We would however expect that effects similar to inertia start playing a role when the delay time approaches the relaxation time of the propulsion mechanism, where the propulsion direction in the previous time step influences the particle velocity in the next time step and therefore alters the dynamics of the oscillation. We have added more information about the experimental time scales in the Methods section 7.1.

R2.4 The density or the average distance in each realization needs to be clarified since it is also a controlled parameter. The definition of clusters uses the criterium of 15 microns. Particles are not quite close considering the radius of the particles is 3 microns. Any reason for choosing this number?

Reply We fully agree that the particle density is an important parameter and in the revised version, we have now added this information to the corresponding figure captions. Regarding the definition of clusters, we first implemented a repulsion algorithm to prevent particles from approaching too closely, as direct contact would cause them to stick together and become unresponsive. The particle distribution around the target demonstrates that they can approach up to about $2.0\ \mu\text{m}$ from the target, which imposes a minimum center-to-center separation of about $10\ \mu\text{m}$ (particle radius is $3\ \mu\text{m}$). This value was therefore used in the repulsion algorithm to prevent sticking. Given the initialization density of $0.003\ \mu\text{m}^{-2}$, for weak or no clustering ($\Gamma \simeq 0.0$), the clustering coefficient is expected to remain low, as there is on average only one particle per box of side length $18\ \mu\text{m}$. Therefore a choice of $15\ \mu\text{m}$ is appropriate to identify clusters. We have clarified this point in the figure captions and the Supplementary Information.

R2.5 It seems that there is a small oscillation flow between left bottom to right up initially in the experiment part of Video 2. Is it true (and why?) or just a visual misperception?

Reply The small global oscillatory drift of ABPs originates from a slight misalignment between the laser focus and the particle center. Because unavoidable mechanical instabilities occur on timescales of hours, we carry out periodic calibrations during the experiments to minimize global drifts. This procedure results in the minor oscillatory global motion correctly noted by the referee. Importantly, this effect does not influence the relative particle motion, and in particular not the swarming behavior reported in our study, as confirmed by the excellent agreement with the numerical (drift-free) data. We have added a clarifying note to the Methods section.

R2.6 There is not experimental counterpart of the mini state loops shown as shown in Fig. 5d

of the simulation results. How is the experimental one?

Reply We did indeed run experiments where we applied the same fast switching of the coupling parameter as in the simulations to reproduce the mini state loops. Each run of these experiments took much longer than the experiments with static interaction (which already took around one hour for each run), so we could switch back and forth between the different coupling parameters multiple times. It is very demanding to control the experimental conditions precisely over such long times and mitigate the inevitable instabilities that lead to the drifts the reviewer has also mentioned in the previous question. Additionally, switching the coupling parameter during the measurement, which also determines the motility of the swarmalators, complicated the periodic re-calibrations during the measurements. For these reasons, we were not able to maintain stable experimental conditions with the high precision necessary to observe the mini state loops.

Reviewer 3

The manuscript presents experimental work on the collective behavior of active Brownian particles that simultaneously exhibit synchronized rotation and translational motion in a coupled fashion, corroborated with numerical simulations and theoretical modelling. A several hundred colloidal particles perform orbiting motion around the reference position of each, steered by a scanning laser with a feedback loop. The reference positions are also moved to induce translational motion of the particles controlled by a coupling parameter (Γ); the reference position of each particle decays toward its average position in the short past for $\Gamma > 0$, while it is moved away from the average position for $\Gamma < 0$.

When the reference positions are fixed on a hexagonal lattice, the particles exhibit synchronized rotation with the average degree of synchronization negatively correlated with the lattice spacing. For motile reference positions, in-phase synchronized pairs tend to attract each other, while anti-phase pairs repel each other. Simulations based on the squirmer model with short-range lubrication interactions reproduce the experimental observation qualitatively, if the swimmers are of the puller-type. For pusher-type swimmers, the correlation between the displacement and degree of synchronization behaves differently.

The authors also introduce a phenomenological swarmalator model with a long-range Kuramoto-type phase coupling and forces that depend on the phase difference between particles. The coupling parameters K for the phase and J for the repulsion/attraction control the collective phase behavior, which is qualitatively consistent with the experimental results for positive (synchronizing) K .

Furthermore, (i) time-varying interactions are realized in the experiment, and a fast switching between positive and negative Γ caused bistable collective states; (ii) lateral forces that are perpendicular to the axis connecting two particles causes global rotation of a cluster in the simulations. The manuscript reports on a state-of-the-art experiment that provides the first controllable realization of a swarmalator system. The results are highly interesting and represent a significant and novel contribution to the emerging field at the intersection of nonlinear science and active matter physics. The qualitative agreement between the experimental results and the numerical simulations is impressive given the system's complexity. Therefore, I can recommend this manuscript for publication in Nature Communications provided the authors can satisfactorily address the following points.

We are grateful to the referee for the careful reading of our manuscript, and for the recommendation of publication.

Main comments:

R3.1 The derivation and justification of the phenomenological swarmalator model [Eqs. (3,4)] should be clarified. The authors should provide a more detailed explanation, perhaps in the Supplementary Material, that addresses the following points:

(i) The $1/r^3$ coupling should be associated with the no-slip boundary condition at the substrate, and the coupling parameter K should be related to the distance from the substrate and other parameters.

(ii) There appears to be a contradiction between the squirmer model, which is based on the Oseen tensor for bulk fluid, and the $1/r^3$ coupling in the swarmalator model, which is associated with hydrodynamic interactions near a no-slip boundary. The authors should elaborate on the connection between these two descriptions or justify the use of the $1/r^3$ form in their phenomenological

model. (iii) The approximation for the longitudinal distance d_{\parallel} needs justification by experimental data. It is unclear how this quantity is related to the cosine term in Eq.(4).

Reply We agree that the justification for the phenomenological swarming model requires further clarification. In the revised manuscript, we have added a detailed explanation in the supplementary information, complemented by simulations, to better motivate the form employed. We would also like to take this opportunity to clarify several related aspects:

- (i) The long-range behavior of light-activated colloids can be modeled as that of a (nearly) neutral squirmer, exhibiting a characteristic $1/r^3$ decay in the bulk, arising from the source dipole. While this has the same distance dependence the hydrodynamic interactions of a pulled bead (force monopole) near a no-slip substrate (Oseen tensor), the underlying physics is fundamentally different: the particles in our study are force-free swimmers, and the $1/r^3$ long-range coupling arises from the squirmer flow fields rather than from any net external force. At short distances however, lubrication forces may effectively act as “pulling” or “pushing” forces, but their exact functional form is not known. To account for these interactions, we implemented in our model a power-law decay with an exponential cutoff at longer distances, with the squirmer radius as the characteristic length scale. This ensures that short-range effects are incorporated without altering the intrinsic long-range coupling, which remains dictated by the squirmer model. Consequently, the resulting long-range interactions in our phenomenological swarming model are consistent with the $1/r^3$ scaling, even though the mechanism differs from the $1/r^3$ interactions of externally actuated particles near a no-slip boundary. There is therefore no contradiction between the squirmer-based description and the form of the long-range coupling used in the phenomenological model. We have now clarified this aspect.
- (ii) The coupling constant K depends on the strength of the underlying hydrodynamic interactions, such as lubrication forces and the source dipole strength s . In the phenomenological model, this dependence is approximated by treating K as an effective constant, which we can vary to explore new phases that are not experimentally accessible – like anti-synchronous phases.
- (iii) The form of d_{\parallel} is directly motivated by experimental data (Fig. 3e), where $d_{\parallel} \sim \sin(\pi\sigma_{ij})$ and $\sigma_{ij} \sim \cos \Delta\theta_{ij}$. We have now added a section in the SI that explains the derivation of equation (4) in more detail to avoid any potential confusion.

R3.2 The phase diagram presented in Fig. 4 is currently depicted as a collection of representative snapshots. To make this a quantitative and more rigorous phase diagram, the authors should define and apply quantitative order parameters to clearly delineate the boundaries between the different collective states (e.g., static synchronous, rotating cluster, async).

Reply We thank the referee for this valuable suggestion. We agree that our previous depiction was primarily qualitative and did not employ quantitative order parameters to clearly delineate the boundaries between the different collective states. In response, we have now constructed an order-parameter space—based on synchronization, cluster size, and cluster asphericity—that enables us

to identify and distinguish the various swarmalator phases. Our analysis shows that the data separate clearly into four phases, corresponding to the representative states presented previously. However, since this analysis is somewhat technical and does not reveal any additional phases beyond those already shown in Fig. 4, we have included it in the Supplementary Information.

R3.3 The emergence of global cluster rotation from lateral forces [Fig. 5g,h] is an intriguing result that requires further explanation. The authors should clarify: (i) What parameters control the magnitude and speed of this global rotation? (ii) How is the direction of the collective rotation (e.g., clockwise or counter-clockwise) determined by the properties or chirality of the individual particles.

Reply The rotation speed is found to depend on the cluster size (as shown in Fig. 5h), as well as on the strength of hydrodynamic interactions and cluster synchronization. Since hydrodynamic interactions modify the trajectory, which couples to the swarmalator motion through Γ , we expect the rotation speed to increase with this parameter. We have now performed additional simulations to verify this, and the corresponding results have been included in the main text.

The rotation direction is determined by the emergent final chirality of the individual swarmalators within the cluster – i.e. the adopted collective rotational state. The individual particles themselves do not possess an intrinsic chirality; they are initially randomized between clockwise and counter-clockwise states. However, noise (which facilitates occasional flips in rotation direction) together with hydrodynamic interactions leading to synchronization causes rotational symmetry breaking, so that all swarmalators adopt the same rotational sense (either clockwise or counterclockwise). The cluster then rotates in the same sense, i.e clusters of CW/ACW moving particles rotates CW/ACW. We have now added this explanation regarding the cluster rotation and supported it with additional data in the Supplementary Information.

Minor issues:

R3.4 To place their findings in a broader context, the authors might consider discussing the observed global cluster rotation in relation to similar phenomena in biological systems, such as the collective rotation of starfish embryos [26].

Reply We have added a remark to this system in the manuscript where we first mention the observation of the rotating clusters.

R3.5 Optical feedback control to steer rotation of colloidal particles and switch synchronization patterns was studied in: A. Maestro et al., Commun. Physics 1, 28 (2018). The paper seems more relevant to the present work than the other quoted examples of anti-phase hydrodynamic synchronization.

Reply We thank the referee for pointing out this publication. We have added it as a reference to our manuscript.

R3.6 In page 5, left column, “Figure 3d presents the average longitudinal displacement ...”,it should refer to Fig. 3e.

Reply Corrected.

R3.7 In Fig. S5, caption, “the data is shows” might be a typo.

Reply Corrected.